# Determining seropositivity—A review of approaches to define population seroprevalence when using multiplex bead assays to assess burden of tropical diseases

YuYen Chan[1]*, Kimberly Fornace[2], Lindsey Wu[1], Benjamin F. Arnold[3], Jeffrey W. Priest[4], Diana L. Martin[5], Michelle A. Chang[5], Jackie Cook[6], Gillian Stresman[1], Chris Drakeley[1]

1 Department of Infection Biology, London School of Hygiene and Tropical Medicine, London, United Kingdom, 2 Department of Disease Control, London School of Hygiene and Tropical Medicine, London, United Kingdom, 3 Francis I. Proctor Foundation, University of California, San Francisco, San Francisco, California, United States of America, 4 Division of Foodborne, Waterborne, and Environmental Diseases, United States Centers for Disease Control and Prevention, Atlanta, Georgia, United States of America, 5 Division of Parasitic Diseases and Malaria, United States Centers for Disease Control and Prevention, Atlanta, Georgia, United States of America, 6 MRC International Statistics and Epidemiology Group, Department of Infectious Disease Epidemiology, London School of Hygiene and Tropical Medicine, London, United Kingdom

* yuyen.chan@lshtm.ac.uk

**Data Availability Statement:** All relevant data are within the manuscript and its Supporting Information files.

## Abstract

### Background

Serological surveys with multiplex bead assays can be used to assess seroprevalence to multiple pathogens simultaneously. However, multiple methods have been used to generate cut-off values for seropositivity and these may lead to inconsistent interpretation of results. A literature review was conducted to describe the methods used to determine cut-off values for data generated by multiplex bead assays.

### Methodology/Principal findings

A search was conducted in PubMed that included articles published from January 2010 to January 2020, and 308 relevant articles were identified that included the terms "serology", "cut-offs", and "multiplex bead assays". After application of exclusion of articles not relevant to neglected tropical diseases (NTD), vaccine preventable diseases (VPD), or malaria, 55 articles were examined based on their relevance to NTD or VPD. The most frequently applied approaches to determine seropositivity included the use of presumed unexposed populations, mixture models, receiver operating curves (ROC), and international standards. Other methods included the use of quantiles, pre-exposed endemic cohorts, and visual inflection points.

### Conclusions/Significance

For disease control programmes, seropositivity is a practical and easily interpretable health metric but determining appropriate cut-offs for positivity can be challenging. Considerations

**Funding:** The author(s) received no specific funding for this work.

**Competing interests:** The authors have declared that no competing interests exist.

for optimal cut-off approaches should include factors such as methods recommended by previous research, transmission dynamics, and the immunological backgrounds of the population. In the absence of international standards for estimating seropositivity in a population, the use of consistent methods that align with individual disease epidemiological data will improve comparability between settings and enable the assessment of changes over time.

## Author summary

Serological surveys can provide information regarding population-level disease exposure by assessing immune responses created during infection. Multiplex bead assays (MBAs) allow for an integrated serological platform to monitor antibody responses to multiple pathogens concurrently. As programs adopt integrated disease control strategies, MBAs are especially advantageous since many of these diseases may be present in the same population and antibodies against all pathogens of interest can be detected simultaneously from a single blood sample. Interpreting serological data in a programmatic context typically involves classifying individuals as seronegative or seropositive using a 'cut-off', whereby anyone with a response above the defined threshold is considered to be seropositive. Although studies increasingly test blood samples with MBAs, published studies have applied different methods of determining seropositivity cut-offs, making results difficult to compare across settings and over time. The lack of harmonized methods for defining seropositivity is due to the absence of international standards, pathogen biology, or assay-specific methods that may impact resulting data. This review highlights the need for a standardized approach for which cut-off methods to use per pathogen when applied to integrated disease surveillance using platforms such as MBAs.

## Introduction

Neglected tropical diseases (NTDs) and vaccine preventable diseases (VPDs) cause a significant burden on populations in developing countries, and effective surveillance plays an important role in the control and elimination of these diseases. Despite the geographical overlap of co-endemic tropical infections in many regions of the world, surveillance efforts have often focused on separate diseases [1]. Integrated approaches to controlling tropical diseases have been implemented in some programmatic settings. However, asymptomatic infections, poor health seeking behaviour, long latency periods, and inconsistent reporting of cases make effective monitoring difficult when relying on passive case detection alone [2].

Serological surveys can be highly informative when assessing the prevalence of diseases or vaccine coverage within a population [3], since antibodies can be used to detect asymptomatic infection and historical exposure to natural infection or a vaccine [1,4]. As integrated approaches to the management of NTDs are being adopted, multiplex bead assays (MBAs) provide a platform to monitor exposure to multiple pathogens from a single blood sample [5]. MBAs typically measure antibody response in median fluorescence intensity (MFI), which is proportional to the levels of antigen-specific antibodies (most commonly IgG) in the blood [6]. Cross-sectional and longitudinal serological data have been used in various public health settings, including evaluating mass drug administration (MDA) campaigns[7,8], assessing changes in population level exposure [9], monitoring transmission patterns [10,11], assessing

the impact of vaccine program coverage [12], and determining prevalence thresholds for confirming disease elimination [7,13].

While serological surveys using MBA provide efficient and cost-effective benefits to integrated pathogen monitoring, a challenge remains in data interpretation. In analysis, the MFI values are often used to estimate the seroprevalence to a particular antigen through calculated or arbitrary cut-off values to define seronegative and seropositive populations. In some cases, higher MFI values are assumed likely to represent more recent or repeated exposure [14,15]. Prior knowledge of specific antibody titres and associated kinetics would be helpful in more accurately interpreting the data. For this review we consider 'seropositive' as a general term that could represent either current or previous infection or vaccination, without interpretation pertaining to specific antibody kinetics or longevity.

The use of a binary seropositivity endpoint allows translation of continuous assay-specific MFI values into a common epidemiologic metric: seroprevalence. Different approaches have been used to define a seropositive response, though the rationale or implication of the method choice is rarely made clear. The choice of approaches used are likely the result of a standard laboratory approach, adopting methods applied in previous studies, or simply ease of use. Antibody responses to different pathogens are intrinsically diverse, making it plausible that specific cut-off methods are better suited for specific antigens or situations. Understanding current approaches used for determining seropositivity is a crucial step in developing standardised methods, ensuring appropriate interpretation of the data to support more robust programmatic decision-making. To address this evidence gap, a literature review was performed of existing methods for determining cut-off values for the assessment of seroprevalence for NTDs and VPDs with MBA.

## Methods

### Review of literature

We conducted a literature review on PubMed for articles published between January 1, 2010 and January 31, 2020. Search terms included "multiplex bead assays" + "serology" + "cut-off". Studies were excluded if they were not in a published journal (e.g. clinical case reports or conference abstracts), published prior to 2010, did not include serological targets for NTDs, malaria or VPDs, or used serological tools specifically for clinical diagnosis. In total, the initial search identified 308 articles of which 253 articles were not included based on title and abstract. Fifty-five articles met the inclusion criteria for full screening which included serology, NTDs (as defined by the World Health Organization and/or PloS NTD lists)[16], malaria, VPDs in tropical regions, and the use of MBA (Fig 1). Articles were then selected if they described cut-off methods using data based on quantitative antibody levels from MBA platform for the application to seroepidemiology.

## Results

### Literature review—Applied methods of determining seropositivity

Eight cut-off approaches were identified based on literature reviewed, with seven methods being applied that provided valid cut-off values (Table 1). A list of all the articles reviewed using each method, antigens within the study, and population origins are listed in S1 Table. Examples of applied methods in different public health settings (Table 1) and the advantages and disadvantages of the different methods (Table 2) are described below.

**Presumed unexposed.** Cut-off values can be determined by a population that has no expected exposure to the pathogen of interest. Depending on the pathogen, these populations

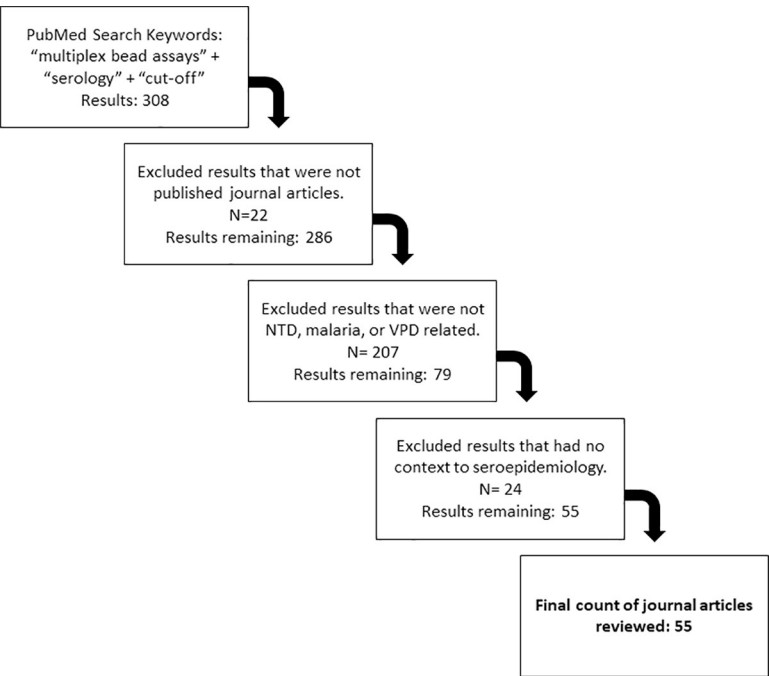

**Fig 1. Flowchart of article selection for inclusion in literature review.** The PubMed search identified 308 articles with 253 being excluded because they did not meet the inclusion criteria. After review, 55 articles were retained for analysis.

are typically selected based on self-reported claims of no travel or no recent travel history to endemic countries. For NTDs, seronegative populations have been chosen from non-endemic regions, including the United States, Sweden, and Japan [6,21,23,29]. When applying presumed unexposed populations to define cut-offs, a pre-specified number of standard deviations (usually three) above the mean of the MFI values with background subtracted (MFI-bg) in the presumed unexposed population are used. Any result above that MFI-bg value is considered as seropositive, and the number of standard deviations used may depend on stringency of identifying seropositives. Use of presumed unexposed populations to determine cut-off values provides a viable option where a large majority of the study population is exposed due to high transmission. In such settings, an endemic seronegative population, as required by other commonly used approaches described below, may be difficult to identify due to small numbers.

However, there are several potential sources of bias to using a presumed unexposed population to derive seropositivity. Cut-off values from presumed unexposed populations run the risk of bias as the immunological exposure of the populations being compared may not accurately represent the immunological history of the sample population. This could be due to factors such as genetic differences affecting immune responses, age differences between presumed unexposed and study population, nutritional status, and/or co-infections of multiple diseases [30–32]. As a result, cut-off values defined by presumed unexposed populations may be artificially low, leading to inflated prevalence estimates. Moreover, differentiating between active and historical infections may not be captured by an overly sensitive cut-off [21]. Conversely, while presumed unexposed populations by definition have no exposure to the infection being monitored, some individuals may have had unknown contact with the pathogen of interest or cross-reactive pathogens that may generate elevated cut-off values if not excluded.

**Table 1. Examples of several applied cut-off approaches using MBAs in various settings for NTDs and VPDs.** MBA used, location, cut-off approach, and the goal of the program are provided in this table to demonstrate the application of cut-off methods in different settings. Where, N refers to the number of studies employing the method and SD refers to standard deviations.

| Disease | Location | Additional cut-off details | Goal (study Ref) |
|---|---|---|---|
| **Presumed unexposed (N = 23)** | | | |
| Dengue | Haiti | United States/mean +3SD | Disease surveillance [9] |
| Lymphatic Filariasis | Mali<br>Haiti | | Disease recrudescence [17]<br>Disease monitoring after MDA [18] |
| Trachoma | Haiti | | Examining MBA as a monitoring tool [19] |
| Amoebiasis | Haiti | | Disease dissemination [20] |
| Leishmania | Kenya | Japan/ mean +3SD | Application of multiplex assays [21] |
| **Receiver Operating Curves (N = 15)** | | | |
| Yaws | Ghana | Clinically confirmed negatives and positives. | Evaluate antibody response in MBA [4] |
| Measles | Kenya | Gold standard lab technique confirmation. | Assess schistosomiasis impact on vaccine preventable diseases [22] |
| Strongyloidiasis | Cambodia | Presumed unexposed population used for ROC curve | Application of an integrated, multiple disease survey [23] |
| **Mixture Models (N = 14)** | | | |
| Lymphatic Filariasis | Kenya | Mean of negative component + 3SD | Validation of MBA to lymphatic filariasis [21] |
| Chikungunya | Haiti | Mean of negative component + 2SD | Estimate exposure [24] |
| **International Standards (N = 6)** | | | |
| Tetanus | Cambodia<br>Tanzania | >100 MFI units = 0.01 IU/ml = seroprotective | Monitor progress of elimination [12]<br>Assess immunity gaps [25] |
| **Quantile (N = 1)** | | | |
| Influenza | Vietnam | No distinct cut-off, use of antibody titres | Estimating population-level antibodies [26] |
| **Visual Inflection Point (N = 1)** | | | |
| Trachoma | Laos<br>Uganda<br>Gambia | Impartial (independent) individuals to determine cut-off | Defining seropositivity thresholds for elimination programs [27] |
| **Pre-exposed Endemic Cohort (N = 1)** | | | |
| Giardiasis<br>Cryptosporidiosis<br>Amoebiasis<br>Salmonellosis<br>E. coli<br>Norovirus<br>Cholera<br>Campylobacteriosis | Haiti<br>Kenya<br>Tanzania | Longitudinal cohort | Understanding force of transmission among children through seroconversion rates [28] |

**Receiver operating characteristic (ROC) curve.**   ROC curves can be used to generate cut-offs by plotting the true positive rate (sensitivity) against the false positive rate (specificity) [33]. The optimal cut-off is considered the value that provides the best discrimination between the true seronegative and seropositive populations, or a cut-off that gives equal weight to sensitivity and specificity [34]. Studies have considered presumed unexposed populations, as defined above, as the true negative population, while true positive populations have been considered as those being either a clinically confirmed case or according to established laboratory gold standards for that pathogen [4,8].

A method with perfect discrimination creates an ideal cut-off between the two populations with no overlap [35], however, it can be rare to observe such separation in the general population (Table 2). Accurate ROC curves to define cut-offs rely on the availability of true negative and true positive reference-populations which are seldom available in practice. Additionally, the reference population used to delineate true positive/negative individuals may also bias results, similar to the disadvantages mentioned for presumed unexposed populations [36].

**Table 2. Summary of Advantages and Disadvantages of different seropositivity cut-off methods.**

| Cut-off Method | Advantages | Disadvantages |
|---|---|---|
| Unexposed or presumed unexposed population | - Known seronegative population<br>- Can be used with other classification methods that require a true seronegative population | - Cut-offs may not reflect true immunity of target population, leading to potential misclassification<br>- Requires obtaining a presumed unexposed population<br>- Only appropriate for certain diseases which are absent in the population from where negatives are selected<br>- Potential for cross reactivity |
| Mixture Model | -Generates cut-off using statistical modelling without external samples needed<br>-Determines an endemic, seronegative population within sample | - May not be appropriate in very high or very low transmission settings<br>- Possibility of an indeterminate range of overlapping seronegative and seropositive individuals |
| ROC Curve | - Robust cut-off generated from true positives and true negatives | - Often requires "gold standard" confirmation of positive and negatives |
| International Standards | - Provided by WHO<br>- Universal method of categorizing seropositivity to enable standardization across assays and laboratories | -Fixed cut-off values may not accurately capture differences in natural and vaccinated responses due to its diagnostic purpose.<br>-Not available for many NTDs. |
| Quantiles | -Visual distribution of MFI intensities and allows for comparison of means | -Choice of which quantiles to use that accurately reflects serostatus must be determined by investigator |
| Visual Inflection Point | -Simple method | -Arbitrarily decided by investigator<br>-May need a statistical method to confirm<br>- Potential for poor reproducibility across settings |
| Pre-exposed endemic cohort | -Provides a presumed seronegative population from the population of interest | -Requires longitudinal data and following individuals who were disease free and later developed disease.<br>-Using MFI values of children may not accurately represent MFI values in adults |

**Finite mixture model.**    A mixture model is a probabilistic model that assumes the presence of at least two normally distributed subpopulations, or components, within the sample population [37]. These components represent underlying populations of varying antibody responses [21,38,39]. The negative population are assumed those within the lowest distribution of MFI values. The cut-off value can be determined using estimated parameters (i.e. mean and standard deviation) of the lowest component specified by the mixture model [40]. Commonly, the cut-off value is then calculated similar to the presumed unexposed approach, using the mean plus a pre-defined number of standard deviations [21,36,41]. Alternatively, mixture models can use joint probabilities of classifying individuals with specific antibody levels as either seropositive or seronegative to specify appropriate cut-off values [39].

Mixture models can, theoretically, provide cut-off values that more closely resemble the target population immunity with a distribution of MFI values representing seronegative individuals within the target population. This is advantageous because baseline seronegative antibody concentrations have been shown to differ between populations due to transmission history of the pathogen of interest, circulating co-infections, and any population-specific genetic factors [21]. Additionally, multiple component mixture models have the potential to identify exposure history. For example, in a mixture model with three components, the lowest component could be considered as seronegative, the middle component as an "indefinite/borderline" or past exposure history group, and the highest component as seropositive or recent exposure group [21,42]. However, the choice of how many components is also a practical challenge for different pathogens and may rely on understanding antigen-specific immunological response. Mixture models can also be fitted to different distributions depending on the pattern of responses of the pathogen of interest, such as in the case of VPDs and distinguishing between stronger antibody responses in naturally infected compared to vaccinated individuals [43,44].

Mixture models may not be appropriate in areas of high transmission or very low transmission [11]. When only one component is observed (e.g. everyone is exposed or unexposed) or when components have significant overlap (e.g. population with large portion of historical exposure or have received treatment), it becomes difficult to identify a reliable cut-off and classify individuals as seronegative or seropositive based on probabilities [21]. Moreover, choice of distribution for fitting mixture model and resulting cut-offs may be rejected if they do not agree with components upon visual inspection and investigator judgment. Co-circulating pathogens that may result in cross-reactivity of antibody response to the antigens being assayed can also be difficult to separate using mixture models [45].

**International standards and units.**    International Standards or International Reference Materials of the World Health Organization (WHO) are used as a simple method for a uniform classification system. This allows comparison of biological targets, such as vaccine induced antibodies, across populations using pre-set cut-off values [46]. This approach requires standard reagents to generate an assay-specific standard cut-off for each of the different antigenic targets which can then be applied consistently across all settings. The main advantage includes the facilitation of between-setting comparisons. However, occasional pre-set international standard cut-off values have been found to overestimate the size of the seronegative population [47] or to classify individuals to incorrect serostatus groups [48]. This could be related to the fact that international standards are decided *a priori* and without context to the populations of interest. Therefore, any potential biases when applying the standard due to population specific genetics are not accounted for, unless they were developed using populations from all endemic countries. Moreover, the international standards may have been developed for specific applications, such as providing a clinical endpoint, and may be less suitable in a seroepidemiological context [37]. For example, international standards for rubella have been found occasionally to overlook potential immunity, due to high cut-offs set by manufacturer assays to avoid false negatives [37].

**Quantiles.**    Cut-offs can be determined through rank statistics that partition MFI values into quantiles of equal probabilities. Quantiles have been used outside the context of NTDs, such as understanding viral loads in influenza [26]. Theoretically, higher quantiles could be interpreted as seropositive, while lower quantiles would be interpreted as seronegative. The partitions of quantiles may furthermore represent different levels of seropositivity, such as populations of non-exposure, of historical exposure, repeated exposure resulting in 'boosting' of antibodies, or populations of active or recent infections. Quantiles require the analyst to subjectively, or based on biological and/or clinical knowledge, choose the number of quantiles for the analysis and then to specify which quantiles are seropositive or seronegative. Additionally, they may assume homogeneity of exposure in quantile groups that could lead to inaccurate estimations [49].

**Visual inflection point (VIP).**    A single study looked at using crude cut-offs determined by visually examining inflection points within MFI distributions in graphs. Migchelsen et al., in exploring options for determining trachoma cut-offs, did a convenience sample of impartial individuals to visually inspect data curves to determine an inflection point [27]. The final cut-off was considered to be the average of values reported by the participants. The mean reported cut-off values were similar to cut-offs from the mixtures models as applied to the same dataset [27]. Moreover, the process is more straightforward and intuitive compared to the mixture model approach.

Use of VIP relies on pattern recognition to subjectively generate cut off values, and inflection points may be biased based on groups of individuals asked. In addition, VIP should ideally use impartial participants and mask antigens to reduce bias. Sampling more individuals to determine the inflection point may improve the precision of the estimates of VIP, but

recruiting a large number of participants can be time-consuming and challenging in certain situations. With this method there are problems with reproducibility, accuracy is likely associated with the degree of separation between the negative and positive distributions.

**Pre-exposed endemic cohort.**   While serological assays are frequently cross-sectional, longitudinal surveys that have obtained serological data before and after infection can create a cut-off based on the change of MFI values before compared to after exposure. Arnold et al. have explored this cut-off method (termed "presumed unexposed" within their study) for enteric pathogen antibody responses among children from Kenya and Haiti [28]. The resulting cut-offs were comparable to both mixture models and presumed unexposed referent populations, but this method also enabled estimation of cut-offs for particularly high-transmission pathogens where other methods failed. In high transmission settings, fitting mixture models can be challenging in the case where distinct components are not present (see mixture model section above), while cut-offs of presumed unexposed from may not reflect immunological background of study population (see presumed unexposed section). A negative population to use for cut-off determination was generated from MFI values of <1-year-olds who later seroconverted (based on a conservative +2 increase on a $\log_{10}$ scale or a 100-fold increase in MFI). The cut-off was determined by taking the mean of the distribution of measurements before these <1-year-old children seroconverted and then adding three standard deviations.

Identifying a pre-exposed endemic cohort population using measurements from individuals who subsequently seroconvert may be useful for longitudinal studies that have collected data on individuals prior to a point change to seroconversion or infection status. However, using MFI values of unexposed infants may not represent the true seronegative MFI values in the adult population due to inherent differences in the immature and mature immune systems. Maternal antibodies may also be present in infants, leading to potentially higher responses in infants that reflects the exposure history of the mother not of the child. The choice of antibody level increase required to identify "pre-exposed endemic cohort" is a qualitative decision, and so accompanying sensitivity analyses of alternate increases could prove useful [28]. Additionally, longitudinal monitoring may not be logistically feasible for many surveillance programs administering cross-sectional surveys.

## Discussion

As programs implement integrated approaches to controlling infectious diseases, effective monitoring is crucial. Serological MBAs provide a convenient method for understanding the population-level burden for multiple diseases simultaneously [50]. This is particularly relevant for those pathogens with long latency periods or with symptoms not sufficiently acute to prompt care-seeking. MBAs can also generate data at a comparatively low cost [1], making it an efficient tool for integrated surveillance of tropical and vaccine preventable diseases. Assessing disease burden through seropositivity is valuable and a more programmatically interpretable metric compared to the continuous MFI values. Additionally, assay and differences in bead coupling concentrations or methods between studies will lead to variability in overall magnitude of antibody levels measured, making the direct comparisons of MFI values almost impossible without appropriate assay standards or a standard metric, such as seropositivity. However, use of seropositivity requires careful consideration of how to define appropriate cut-off values that can meaningfully identify exposed individuals and those with disease burden according to public health programmatic guidelines.

This review highlights several approaches for determining seropositivity cut-offs. The most frequently used approaches were presumed-negative populations, ROC curves, mixture models, and international standards. Other approaches included quantiles, pre-exposed endemic

cohort, and visual inflection points. Each method has its respective advantages and disadvantages. For all methods that rely on external samples, such as presumed unexposed population or ROC curve, it is important to acknowledge that antigen-bead coupling efficacy may differ between bead batches and, if not run on the same bead set, potential differences in cut-off values may be observed. In addition, instrumentation differences may impact the stability of the cut-off values. Under these circumstances, additional adjustments to the MFI values maybe required for appropriate comparisons. Additional factors important to consider in identifying the most appropriate method for any given context include: the availability of confirmed seronegative and seropositive populations that are necessary for methods such as ROC and presumed unexposed; use-case scenarios based on program targets or goals; transmission intensity factors that impact the seronegative and seropositive distributions for methods that assume sub-populations; methods previously used in similar settings and diseases; and complexities in certain pathogen-host immunobiology that queries the suitability of strict cut-offs (Box 1).

## Box 1: Summary of factors to consider and complexities in choosing cut-off methods

*Availability of Seropositive and Seronegative Populations*

The availability of expected true seropositive and seronegative populations through screening, clinical confirmation, populations from countries without transmission, or gold standard laboratory techniques justifies the use of presumed unexposed approach, ROC curves, and other supervised classification methods. Additionally, if there are large differences in endemicity within a country, populations from low or non-transmission areas could serve as a seronegative population. Having the presence of seropositive and seronegative populations does not the exclude using other methods, however. Additionally, precision, quality, and interpretation of cut-off values are impacted by a variety factors that should be taken into consideration along with the method of determining cut-offs.

*Large sample sizes*

As with many statistical methods, larger sample sizes allow for a better estimation of the target population, improving both sensitivity and specificity. Additionally, certain cut-off methods, such as mixture models, can be achieved with larger sample sizes. Smaller sample sizes may require fitting different distributions [56].

*Use-case scenarios*

Cut off methods can be chosen depending on the goal and design of the study or the program. For example, cut-off methods such as ROC with high sensitivity or specificity may be preferred in the case of assessing program coverage [11, 19]. Cut-off methods such as quantiles or mixture models with several components that can identify multiple levels of seropositivity may be chosen when trying to understand geographic transmission patterns.

*Literature past precedent or international guidelines*

Decision to use a certain method could be influenced by or borrowed from other studies that focus on biologically similar diseases. This also includes international guidelines that provide cut-offs for vaccine preventable diseases. This consideration offers a simple

and convenient rationale to choosing a certain cut-off method given that the cut-off has already been established. Making comparisons between studies using similar antigens may also determine the use of a certain cut-off method.

*Transmission Dynamics*

The justification of using certain cut-off methods may depend on the level of transmission of the pathogens. Mixture models and quantiles are more appropriate in transmission areas where the seropositive and seronegative components have some separation evident in the MFI distributions.

*Complexity of the immunology of host-pathogens interactions*

The use of statistical methods is an attempt to reflect a biological process in terms of exposure or lack of exposure. While statistical methods for cut-offs are important in determining seropositivity, weight should also be placed on understanding the complex immunology of the NTD of interest and the immunological background of the population. Incomplete understanding of serologic response and other immune mechanisms against pathogens of interest may impact interpretation of prevalence estimates generated from cut-offs [47]. For example, population level antibodies due to partial or waning immunity could make it difficult to define a strict cut-off value for seropositive and seronegative groups [11]. It may also be unclear whether responses observed during a chronic infection ever revert to a seronegative state [35]. Therefore, using an indeterminate range or comparing the mean MFIs in these circumstances maybe more appropriate than enforcing a strict cut-off value [59, 60].

*Antigen and antibody dynamics*

Antibody responses are inherently noisy and imposing a strict cut-off may lead to misclassification [61]. Furthermore, antibody longevity may impact seropositivity classification [60, 62]. Coinfections can also be difficult to detect and separate, as certain pathogens with high titres can dominate detection assays [63].

In addition, antibody dynamics in terms of boosting and decay rates post infection should be taken into consideration. For example, as control programs lead to less disease exposure in populations, lower amounts of infection-specific antibodies circulate in the population and are replaced by residual antibody responses [64]. Roscoe et al. noted *S. stercoralis* antibodies decreased over time but remained above cut-off values a year and a half after successful treatment [65]. When determining prevalence estimates with cut-off values, some of these responses may actually be the result of cleared infections with residual antibodies.

Moreover, the dynamics of antibody-antigen interactions within age groups such as children and adults should be considered when interpreting cut-off values as they have been shown to differ [66]. For instance, cut-off values determined from a population of children may not be appropriate for the entire population age range when assessing prevalence of certain pathogens as children's immune systems are predominantly short-lived B-cells, while antigen presentation and helper T-cell function are more developed in the immune systems of adults[11, 67, 68]. Lastly, the inherent nature of antibody classes, such as IgG vs IgM, may be interpreted differently regardless of cut-off method [69].

*Laboratory technique and design*

Although not a focus of this paper, laboratory techniques impact the quality of MFI values. Thus, the generation of good quality cut-off values and resulting prevalence estimates require appropriate assay validations with sufficient quality control protocols [70]. Additionally, cut-off thresholds are dependent on specific coupling conditions [71], and bead consistency is an absolute requirement for the generation of precise cut-off values, regardless of the cut-off determination method.

As more programs implement serological surveillance strategies for neglected tropical disease monitoring, it is possible that new cut-off methods will be developed and applied. Alternatively, other classification methods without a distinct cut-off, such as K-means clustering, aims to separate high dimensional data (i.e. multiple antigenic targets for the same pathogen) into different clusters of MFI values to represent seronegative and seropositive states could be implemented [51,52]. Use of multiple target antigens will increase the likelihood of detecting previous exposure to infection as well as reducing the likelihood of non-reactivity due to sequence variation in single antigenic targets and differential immunogenicity. However, in multi-disease panels, antigens need to be well-defined in order to avoid potential cross-reactivity that could lead to issues of inaccurate or false results due non-specific binding [53]. Furthermore, heterogeneity of individual responses that influence antibody levels apart from pathogen exposure, e.g. nutrition or health conditions, can cause increased immunoglobulin in sera, such as hypergammaglobulinemia [54]. Refining statistical techniques that allow assessment of multiple and/or combinations to generate seroprevalence will also be of benefit and aid in interpretation of data [55,56].

Within our review, there are several limitations. Our search criteria targeted serological cut-offs according to WHO and PLOS definition of NTDs and VPDs, specifically in PubMed and in English. However, there may have other methods to determine serological cut-offs for diseases were not included in this review from other databases and also outside the specific timeframe we examined. Additionally, the search criteria focused only on the term "cut-off", which may have overlooked similarly terminology, such as "threshold" or "inflection point" that could have provided additional cut-off approaches. Our study also reviewed cut-offs primarily from MBA and enzyme-linked immunosorbent assay (ELISA) platforms due to our search criteria and did not include other serological or commercial immunoassays that may have used other approaches. However, any additional methods that we could have identified are unlikely to change the conclusions of this work.

International standards based on a large sample of reference standard sera from individuals in known elimination settings will be needed to define universal cut-offs and make program decisions based on specific levels of seropositivity. This would require procuring sera from clinically confirmed individuals with infection and those without infection from a geographically representative number of endemic countries to ensure sufficient diversity of immunological responses, as were recently done for human African trypanosomiasis [57, 58]. Sera from these candidates would then be characterized by different immunological tools to determine consistent measurements of immunological activity (with context to programmatic use) across all platforms in the form of international units. In the absence of these metrics for NTDS, ROC curves with confirmed positives and negatives from the study population are recommended as they would likely generate the most representative cut-offs that consider immunological and genetic backgrounds of the population. Without control sera mixture models are

recommended as they may provide statistically robust cut-offs when adjusting for transmission intensity by using appropriate distributions and number of components to identify seropositives. In the context of integrated disease surveillance, the recommendation for an appropriate cut-off method to determine seroprevalence should additionally consider the antigen being assessed, the optimal data that is the closest reflection of true population prevalence, and other important factors and complexities that could impact decision of cut-off method listed in Box 1.

## Disclaimer

The findings and conclusions in this report are those of the authors and do not necessarily represent the official position of the Centers for Disease Control and Prevention. Use of trade names is for identification only and does not imply endorsement by the Public Health Service or by the U.S. Department of Health and Human Services.

## Supporting information

**S1 Table. List of Articles Reviewed.**
(DOCX)

## Author Contributions

**Conceptualization:** YuYen Chan, Gillian Stresman, Chris Drakeley.

**Formal analysis:** YuYen Chan.

**Methodology:** YuYen Chan, Kimberly Fornace, Lindsey Wu, Benjamin F. Arnold.

**Supervision:** Gillian Stresman, Chris Drakeley.

**Writing – original draft:** YuYen Chan.

**Writing – review & editing:** Kimberly Fornace, Lindsey Wu, Benjamin F. Arnold, Jeffrey W. Priest, Diana L. Martin, Michelle A. Chang, Jackie Cook, Gillian Stresman, Chris Drakeley.

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
