## [Decision Letter · Decision Letter 0]

7 Dec 2020

Dear Mr Chan,

Thank you very much for submitting your manuscript "Determining Seropositivity - A Review of Approaches to Define Seroprevalence when using Multiplex Bead Assays to Assess Burden of Tropical Diseases" for consideration at PLOS Neglected Tropical Diseases. As with all papers reviewed by the journal, your manuscript was reviewed by members of the editorial board and by several independent reviewers. In light of the reviews (below this email), we would like to invite the resubmission of a significantly-revised version that takes into account the reviewers' comments. 

Two reviewers have suggested that this should be split into two separate papers due to the disparate topics addressed (a systematic review of cut-off determination methods, and a comparison of different methods used in Haiti and Malaysia). 

I suggest that, once the issues identified by the reviewers in both sections have been very thoroughly addressed, the authors may re-submit the systematic review of cut-off determination methods under this manuscript number, while a separate submission should be created for the Haiti and Malaysia data.

We cannot make any decision about publication until we have seen the revised manuscript and your response to the reviewers' comments. Your revised manuscript is also likely to be sent to reviewers for further evaluation.

Sincerely,

Richard Stewart Bradbury, PhD

Associate Editor

Francesca Tamarozzi

Deputy Editor

Two reviewers have suggested that this should be split into two separate papers due to the disparate topics addressed (a systematic review of cut-off determination methods, and a comparison of different methods used in Haiti and Malaysia). 

I suggest that, once the issues identified by the reviewers in both sections have been very thoroughly addressed, the authors may re-submit the systematic review of cut-off determination methods under this manuscript number, while a separate submission should be created for the Haiti and Malaysia data.

Reviewer's Responses to Questions

**Key Review Criteria Required for Acceptance?**

**Methods**

-Are the objectives of the study clearly articulated with a clear testable hypothesis stated?

-Is the study design appropriate to address the stated objectives?

-Is the population clearly described and appropriate for the hypothesis being tested?

-Is the sample size sufficient to ensure adequate power to address the hypothesis being tested?

-Were correct statistical analysis used to support conclusions?

-Are there concerns about ethical or regulatory requirements being met?

Reviewer #1: The paper comprises two separate parts -a systematic review of cut-off detemination methods, and a comparison of different methods used in Haiti and Malaysia. The two parts do not depend on one another or hang together particularly well. 

Please consider separating this into two papers and then each part could be explored more fully. 

The description of the search is confusing - how many more were found from the period between Oct 2018 and Jan 2020 and how many were from ref lists of first included studies? How many studies were in each method/category? 

Supp Table 1 is just a list of papers and Table 1 is very minimal - not clear how many did each method. Some locations that are in papers cited in Supp Table 1 do not appear in Table 1 (papers by Won in American Samoa and Gambia, for example). There may be others. 

What cutoffs did the authors find in each paper with each antigen and method? DId it vary by transmission setting, technical factors or other?

I am not sure what is gained from the papers to generate Table 2. It's a useful summary of potential biases, but could likely have been written without reading any of them.

Reviewer #2: 1.While the authors reference earlier publications, additional brief summaries to describe the assays used including source of antigens and the performance characteristics would be beneficial to the reader when interpreting the results. 

2. Suggest clarifying if the case study results were based on collection of new laboratory data for the sample sets or re-analysis of previously collected serology data for these targets.

Reviewer #3: See attached file

**Results**

-Does the analysis presented match the analysis plan?

-Are the results clearly and completely presented?

-Are the figures (Tables, Images) of sufficient quality for clarity?

Reviewer #1: There are some serious limitations in presentation. The important new results of the paper are in Table 3 and Fig 2. The title of Table 3 is rather obscure, but seems to be comparing 3 methods used in the case studies.

The antibodies used for LF are Wb123 and Bm14 (not Bm14 and Bm33 as given in Abstract where the results appear to refer to Bm14, but numbers are slightly different and some are from each site). For Strongyloides also, abstract is reporting only one site. 

I think in Abstract it would be better to present general relationships between the methods, not just pick a few to report, or mention similarities/differences between the methods/sites. If some are more important and are highlighted, then clearly state which Ag and country being mentioned. 

In Fig 2, axis is stated to be MFI-bg, but isn't bg the background that should be subtracted?

Supplemental Table 1 is in the main text and out of order with Table 4 which I find easier to understand. 

Language not clear sometimes:

line 382 "Prevalence estimates between non-exposed populations... "

I think you mean "Prevalence estimates derived USING THE DIFFERENT CUT-OFF METHODS OF non-exposed populations,... "

Similar issues elsewhere e.g. line 298 where you refer to the method rather than the cutoff derived from a method.

Suggestion - call them methods A, B, C, or NOP, MM, Qtile and refer to them consistently that way e.g. the NOP method-derived cutoff or perhaps just NOP-cutoff.. Please check whole paper for this issue. 

What is meant by lines 386 to 387.. "the choice of cut-off method ... by district were generally consistent". Consistent with what? How are you making the choice?

Table 4 legend: I dont think these are rankings? line 400

If the point is that they rank in same order, maybe colour in the table cells, or a chart would help, as well as putting Supp Table 1 after Table 4. Right now it is hard to see the points made.

Figure 3 is interesting and makes me want to know more about clustering, but the Fig is not very good quality. I cannot see the red mentioned in the legend. What do we infer from this? either method is OK?

In general I did not perceive the results of this work clearly. What method should we be using? Or at least - which method gives consistently high cutoffs and which ones low? The Discussions states some fairly obvious points - e.g. 440-441, lines 458 to 463 that could have been written before this paper and perhaps belong in introduction. We do not need this paper to tell us that cutoffs are important. How does this paper take us further? While the caveats in lines 445 to 447 are true, this is always the case. We should be able to conclude something programmatically (even if provisionally) from the case studies at least, otherwise what is the point of the study? I would have welcomed discussion of the need for site-specific cutoffs which is new information shown in Fig 2 but dismissed as technical issues in the Fig legend. Why not transmssion issues?

For deciding priorities for doing MDA - which cutoff method (or doesn't it matter)?

For investigating clustering - which cutoff method (or doesn't it matter)?

Box 1 is very long and diverts into a lot of side topics. It may be useful but doesn't seem to arise from the data in this paper. Again could be valuable in separate review paper, but is strangely ordered after the case studies.

Reviewer #2: (No Response)

Reviewer #3: See attached file

**Conclusions**

-Are the conclusions supported by the data presented?

-Are the limitations of analysis clearly described?

-Do the authors discuss how these data can be helpful to advance our understanding of the topic under study?

-Is public health relevance addressed?

Reviewer #1: The conclusion needs to be rewritten. Currently it is general, states the obvious and could have been written before the study was done. It does not draw any conclusion from the results, such as how many methods were found, or that there were site-specific differences even with the same method. No clear recommendation is made.

Reviewer #2: (No Response)

Reviewer #3: See attached file

**Editorial and Data Presentation Modifications?**

Reviewer #1: Don't be afraid to draw preliminary conclusions from the findings and make provisional recommendations. 

Make sure to label the antibody tests correctly and describe the different methods and cutoffs in a clear, consistent and simple way. 

Abstract is cherry picking some results right now. Be clear which ones are being highlighted or synthesize better. 

Title - I think it should specify Vaccine Preventable and Neglected Tropical Diseases. 'Tropical Diseases' is too broad. I was expecting malaria to be included. 

Minor point: inappropriate use of the word 'confounders' line 195 . It has a specific meaning in epidemiology. I think here it should be 'potential sources of bias' not confounding.

Reviewer #2: 1. A key concern is the cohesion in the paper between the review and the case study. A recommended solution would to consider segregating this into two separate publications or revise the case study with more focus on a couple of diseases as a an example. 

2. It is unclear why the authors are emphasizing multiplex bead-based serology assays as a theme of the paper. The initial landscape section regarding analysis approaches could be adapted and would likely be applicable to methods used for defining cut-offs for quantitative serology assays regardless of platform. While there are certain characteristics that are specific to bead-based serology assays (such as read-out using MFI), wouldn't the issue of using different methods for defining cut-offs also be applicable to other quantitative serology platforms some of which have also been used to develop multiplex serology assays. Broadening the focus would likely add value as a more comprehensive review article with broader interest to people conducting seroepidemiology studies. 

3. As the article is currently written, I am not really clear on the why they are emphasizing the multiplex component in both the title and the paper. The results and the discussions in the case study focus on the impact of applying these different methods defining cut-offs for individual serologic assays. Perhaps this could be clarified through revision of the case study to focus on 1-2 examples rather focusing 4 diseases.

Reviewer #3: See attached file

**Summary and General Comments**

Reviewer #1: Overall there is useful information in this paper that deserves to be seen, especially from the case studies. But the two parts do not hang together and more could be made of each. 

In the systematic review part, please give more details rather than just a list of who did what method. Make sure list is comprehensive. What cutoffs did they each find? What were the background transmission situations.? What stage was the programme at? Are these antigens/antibodies currently used in decision making? Should they be, now that MBA assays are available? Also, how about comparing with or at least mentioning studies that used mixture models with conventional ELISAs e.g (https://www.ncbi.nlm.nih.gov/pmc/articles/PMC6473238/) 

In the case study part, it is more a question of interpretation. Results are presented but then seem to be dismissed as not applicable or too biased. Perhaps suggest what size of sample needed to use mixture models or other methods. Investigate site-specific cutoff differences further. Name some situations where one or other methods could or should be used, and what consequences might be for under or overestimating the prevalence.

Reviewer #2: The topic of the publication is an important and relevant one given the increasing interest and potential value of using serology and integrated disease surveillance approaches to provide more cost-effective and operationally feasible solutions to conduct population surveys to inform decisions for NTDs and other disease programs. Suggest minor revision with particularly focus on the presentation of the case study.

Reviewer #3: See attached file

PLOS authors have the option to publish the peer review history of their article (what does this mean?). If published, this will include your full peer review and any attached files.

Reviewer #1: No

Reviewer #2: No

Reviewer #3: No
---

## [Decision Letter · Decision Letter 1]

6 Apr 2021

Dear Mr Chan,

Thank you very much for submitting your manuscript "Determining Seropositivity - A Review of Approaches to Define Population Seroprevalence when using Multiplex Bead Assays to Assess Burden of Several Vaccine Preventable and Neglected Tropical Diseases" for consideration at PLOS Neglected Tropical Diseases. As with all papers reviewed by the journal, your manuscript was reviewed by members of the editorial board and by several independent reviewers. The reviewers appreciated the attention to an important topic. Based on the reviews, we are likely to accept this manuscript for publication, providing that you modify the manuscript according to the review recommendations. 

The revised version of this manuscript is much improved, but the reviewers have still identified some areas requiring attention, and a minor revision to address these reviewer comment is indicated.

Sincerely,

Richard Stewart Bradbury, PhD

Associate Editor

Francesca Tamarozzi

Deputy Editor

The revised version of this manuscript is much improved, but the reviewers have still identified some areas requiring attention, and a minor revision to address these reviewer comment is indicated.

Reviewer's Responses to Questions

**Key Review Criteria Required for Acceptance?**

**Methods**

-Are the objectives of the study clearly articulated with a clear testable hypothesis stated?

-Is the study design appropriate to address the stated objectives?

-Is the population clearly described and appropriate for the hypothesis being tested?

-Is the sample size sufficient to ensure adequate power to address the hypothesis being tested?

-Were correct statistical analysis used to support conclusions?

-Are there concerns about ethical or regulatory requirements being met?

Reviewer #1: This is a literature review. The objectives and the search strategy are clearly described now. No statistical analysis is done and no ethical clearance necessary.

Reviewer #3: Line 108-110: the authors said that "Fifty five articles met the inclusion criteria for full

screening which included serology, NTDs (as defined by the World Health Organization)(15), malaria,

VPDs in tropical regions, and the use of MBA (Figure 1). However, they mentioned others pathogens not listed in WHO list. For TABLE 1: The authors have to choose between the NTDs recognized by WHO and NTDs recognized by PLoS NTD (see Hotez et al. PLoS NTD 2020 Jan 30;14(1):e0008001. 

E.g Giardia, Entamoeba, ..etc

Reviewer #4: The authors have adequately addressed most of the previous reviewers questions

**Results**

-Does the analysis presented match the analysis plan?

-Are the results clearly and completely presented?

-Are the figures (Tables, Images) of sufficient quality for clarity?

Reviewer #1: The tables in main document are fine. Table 1 should be placed earlier when it is first mentioned Page 6. The different cutoff determination methods are more clearly described. 

Supplementary Table 1 - the cutoff determination methods are not in same order as in the text, and many papers are repeated. I believe Supplementary Table 1 would be much more useful if revised as follows:

List each paper only once and put it in short form e.g. (ref no) Arnold et al 2019. Provide additional columns at right to show which methods the paper uses (Presumed negative, ROC etc) with tick or Y in the column if they used relevant method. It will then be easier to see which papers used which methods, and which tested multiple methods. Most (all?) papers are referenced in main text or can be added so you can use the same number as reference. Or have a list of refs to supp material. 

Then you can sum the number of papers using each method in last row. 

The Finite Mixture model method is described incorrectly. Lines 168 to 171. It does not use mean and distribution of the 'negative population' since the composition of the negative population is not known. It postulates two or more distinct sub-populations (components) and assesses when probability of a data point/sample being in one or the other is >50% (or other defined value). The text in those lines seems to refer to the Presumed negative population method. Same problem in Table 1 for this method. It is not Mean + 3 SD. The component populations may overlap.

Reviewer #3: Line: 182-184: I could not understand how would this model be able to differentiate between the naturally infected and the vaccinated individuals, is based on the cut-offs as discussed before or comparing to the nature of responses seen during the studies. Because the responses of the population might change due to various factors involved 

TABLE 2. should be revised and linked with the references used in the text describing each cut-off method. Otherwise it is very difficult to follow. For example, for the International Standards, the reference # 32 is cited in the disadvantages but not cited in the text.

Reviewer #4: The presnetationof data is appropriate for the goals of analysis

**Conclusions**

-Are the conclusions supported by the data presented?

-Are the limitations of analysis clearly described?

-Do the authors discuss how these data can be helpful to advance our understanding of the topic under study?

-Is public health relevance addressed?

Reviewer #1: Conclusions are reasonable. Please explain 'past precedents'. It is not clear to me.

There is a paragraph on limitations but may have missed a few - e.g. only papers in PubMed, only in English? Time boundaries?

Reviewer #3: DISCUSSION: 

Line 382: Would you expect any changes that would take place in the cut-offs for the various detection of parasites when they are couped together?

Reviewer #4: The conclusions are supported by the data presented and the authors have acknowledged the limitations of their study design

**Editorial and Data Presentation Modifications?**

Reviewer #1: 1. Apparently malaria IS included (and is not in the categories of Vaccine Preventable (not quite yet) or NTDs. So perhaps revernt to using Tropical Diseases or alternatively just just delete the part of title after Mulitplex Bead Assays

2. Explain past precedents line 36 (what was done before?)

3. line 78 - can you give a citation for use in evaluating MDA programs?

4. Table 1 for VIP-Trachoma - what do you mean by Non-impartial? Partial?

5. Line 238. Maybe a word missing like 'and' before accuracy?

6. Lines 245-247. WHy would this method allow estimation of cutoffs for particularly high transmission pathogens where other methods failed? Can you elaborate and/or give an example?

7. Lines 255 to 256. Wouldn't it be maternal immunity (transfer of antibody from mother to child) that is also a reason not to use infants as 'unexposed' individuals?

8. Discussion Para 2 is very long and covers several topics. Please split into separate paragraphs. E.g. line 286 and 292

9. Lines 323 - not clear why monoclonal antibodies are mentioned. To test for true positives and negs? 

10. In Box 1. Last para but one. Predominantly not predominately

11. Ref 64 word missing in title after Their.

Reviewer #3: ABSTRACT:

Line 29: neglected tropical diseases or vaccine preventable diseases should be abbreviated, NTD and VPD, respectively.

INTRODUCTION: 

Line 71-73: Some individuals might receive treatment for a specific disease and be still antibody positive for a while. How to distinguish dissociate this category and the asymptomatic group. In addition, since most of these NTD are present in developing and Lower-middle-income Countries a non-negligible of people self-medicate themselves.

Line 87-89: So when it is said that all these factors are combined and put under the umbrella of “seropositive”, but they also differ on the basis of the ½ life of antibodies, a little more explanation on how they would be affected would make his more understandable as it seems like a general statement. In addition, it is also good to mention how cross-reactivity would play a significant role when dressing these responses.

RESULTS: 

Line 136: "Any result above that MFI-bg value is considered as seropositive" should read ....above that MFI-bg plus 3 times the value of the standard deviation is considered....

Line 148-150: When dealing with cross reactivity, the authors should elaborate and take in account people displaying polyclonal hypergammaglobulinaemia.

REFERENCES:

Line 381 and 384: be consistent.

Line 433: is missing Journal name and volume.

Line 515: Reference of a book Chapter, not complete.

Reviewer #4: No significant modifications recommended

**Summary and General Comments**

Reviewer #1: Overall this manuscipt is much improved and a useful summary of the possible methods for establishing cutoffs when using multiplex assays.

Reviewer #3: Please see comments above

Reviewer #4: The revised manuscript has largely addressed the points raised by the reviewers, summarises the methodologies applicable to cutoff determination in serological testing and adds insight to the debate on optimal selection of cutoffs for serological bead array-based assays. A missing element in the conclusions are the authors own recommendations for the appropriate methodology in specific settings Box 1 does not add significantly to, or with any more clarity, what is in the results and discussion and appears to be superfluous.

PLOS authors have the option to publish the peer review history of their article (what does this mean?). If published, this will include your full peer review and any attached files.

Reviewer #1: No

Reviewer #3: No

Reviewer #4: No

Figure Files:

Data Requirements:

Reproducibility:

References

---

## [Editor Report · Decision Letter 2]

10 May 2021

Dear Mr Chan,

We are pleased to inform you that your manuscript 'Determining Seropositivity - A Review of Approaches to Define Population Seroprevalence when using Multiplex Bead Assays to Assess Burden of Tropical Diseases' has been provisionally accepted for publication in PLOS Neglected Tropical Diseases.

Best regards,

Richard Stewart Bradbury, PhD

Associate Editor

Francesca Tamarozzi

Deputy Editor

---

## [Editor Report · Acceptance letter]

23 Jun 2021

Dear Mr Chan,

We are delighted to inform you that your manuscript, "Determining Seropositivity - A Review of Approaches to Define Population Seroprevalence when using Multiplex Bead Assays to Assess Burden of Tropical Diseases," has been formally accepted for publication in PLOS Neglected Tropical Diseases.

Best regards,

Shaden Kamhawi

co-Editor-in-Chief

Paul Brindley

co-Editor-in-Chief
